# REVEALING STOCHASTIC NOISE OF POLICY ENTROPY IN REINFORCEMENT LEARNING

## ABSTRACT

On-policy MARL remains attractive under non-stationarity but typically relies on a fixed entropy bonus that conflates useful exploration with stochastic fluctuations from concurrently learning agents. We present *Policy Entropy Manipulation (PEM)*, a simple, drop-in alternative that treats entropy as a noisy measurement to be *denoised* rather than uniformly maximized. PEM applies positive–negative momentum to *entropy gradient* to form a high-pass, variance-controlled signal that preserves persistent exploratory trends while suppressing transient spikes. The method integrates seamlessly with heterogeneous, permutation-based on-policy algorithms, requires no critic or advantage changes, and uses a one-step warm start that recovers the standard entropy objective before momentum takes effect. Across benchmark tasks, PEM yields smoother and more stable learning curves, and improves coordination under heterogeneous observation/action spaces, resulting in stronger generalization than conventional entropy-regularized baselines. Our results indicate that noise-aware control of the entropy channel is an effective and principled way to stabilize exploration in cooperative MARL.

## 1  INTRODUCTION

Multi-Agent Reinforcement Learning (MARL) is a paradigm in which multiple agents learn policies through decentralized decision-making within a shared interactive environment. It constitutes a critical extension of the achievements of single-agent reinforcement learning to a broader spectrum of multi-agent domains. However, the simultaneous updates of each agent's policy induce non-stationarity by continuously shifting the underlying environmental distribution. When coupled with the inherent complexities of Decentralized Partially Observable Markov Decision Processes (Dec-POMDPs) (Oliehoek et al., 2016), this non-stationarity renders the joint pursuit of stable convergence and efficient exploration particularly challenging.

To address these difficulties, on-policy updating method refines policies using data collected exclusively under the current policy, thereby mitigating distribution shift and enabling the explicit enforcement of constraints that guarantee policy improvement. Notably, Trust Region Policy Optimization (TRPO) (Schulman et al., 2015) and Proximal Policy Optimization (PPO) (Schulman et al., 2017) exemplify this principle by constraining excessive policy updates through trust regions or clipping ratios, which enhances training stability.

Although on-policy methods may suffer from inefficiency in terms of data reuse, they retain substantial value in MARL settings characterized by severe non-stationarity, as they preserve consistency between the policy and the data distribution. The development of on-policy extensions for MARL can broadly be categorized into three directions. First, independent PPO (IPPO) represents a straightforward variant in which each agent independently applies PPO by estimating only a local value function (De Witt et al., 2020). Second, on-policy methods in Centralized Training with Decentralized Execution (CTDE) framework (Chen, 2020): approaches such as Multi-Agent Proximal Policy Optimization (MAPPO) combine decentralized actors with a centralized critic to reduce variance induced by partial observability and inter-agent interactions, while employing techniques such as generalized advantage estimation (GAE) and entropy regularization to promote effective exploration (Yu et al., 2022). Third, trust-region adaptations to the multi-agent setting: methods such as HATRPO (Shek et al., 2025) and HAPPO (Zhong et al., 2024) impose agent-wise Kullback–Leibler (KL) constraints or clipping to theoretically guarantee monotonic improvement of the joint policy.

In this paper, we focus on two key challenges: (i) policy entropy, while widely used for exploration, is a noisy measurement that mixes beneficial exploratory signals with detrimental stochastic fluctuations induced by non-stationarity; and (ii) heterogeneous-agent settings exacerbate this issue, as agents operate with different observation and action spaces, making fixed entropy coefficients unstable and hard to tune.

To address these challenges, we discuss the following:

- Policy Entropy Manipulation (PEM), a novel on-policy optimization method that replaces raw entropy with a positive-negative momentum buffer of entropy increments. This mechanism suppresses high-frequency stochastic noise while preserving persistent policy exploration.

- Integration of PEM into the on-policy reinforcement learning framework, showing that it stabilizes training dynamics and improves coordination among multi-agents without additional critic modifications.

Extensive experiments across benchmark tasks demonstrate that PEM yields more stable learning curves, reduces policy thrashing, and achieves stronger generalization compared to existing entropy-regularized baselines.

## 2 RELATED WORK

**Entropy Regularization in Policy Update.** Entropy bonuses are a long-standing device for stabilizing policy optimization and encouraging sufficient exploration. Early policy-gradient and actor–critic methods routinely add an entropy term to the policy objective to avoid premature collapse to near-deterministic policies (Williams, 1992; Mnih et al., 2016). In PPO (Schulman et al., 2017), this takes the form of maximizing a clipped surrogate objective augmented with an entropy bonus, which empirically improves learning stability across a wide range of tasks. Subsequent analyses study how entropy alters the optimization landscape and gradient signals, clarifying when and why it improves performance (Ahmed et al., 2019). In multi-agent settings, MAPPO adopts the same entropy-regularized policy update under centralized training with decentralized execution (Yu et al., 2022). While these approaches treat all entropy as uniformly beneficial, they do not differentiate between *exploratory* entropy and *stochastic noise* arising from non-stationarity.

**Maximum Entropy Reinforcement Learning.** Maximum entropy RL optimizes return plus an entropy term *as the objective*, yielding policies that remain stochastic even at convergence and improving robustness and exploration (Ziebart et al., 2008; 2010). Soft actor–critic (SAC) implements this paradigm with a soft policy iteration scheme and an automatic temperature adjustment that targets a desired entropy level (Haarnoja et al., 2018). While max-ent methods provide strong guarantees and sample efficiency in single-agent, off-policy regimes, directly transplanting them to cooperative multi-agent on-policy PPO-style updates can be nontrivial due to coupled non-stationarity and the need to coordinate exploration across agents. Our work is closer to on-policy MARL (e.g., MAPPO), but differs by treating policy entropy as a *noisy measurement* that should be denoised before being used to steer exploration, rather than always maximized.

**Heterogeneous-Agent Reinforcement Learning.** Most cooperative MARL methods assume homogeneous agents and rely on parameter sharing, which limits applicability when observation or action spaces differ. While algorithms such as IPPO and MAPPO have shown strong empirical performance (De Witt et al., 2020; Yu et al., 2022), they lack theoretical grounding in heterogeneous settings. Recently, Zhong et al. (2024) introduced the Heterogeneous-Agent Reinforcement Learning (HARL) framework, which establishes a principled foundation for heterogeneous MARL. Building on this framework, variants such as HAPPO (Zhong et al., 2024) and HATRPO (Kuba et al., 2021) extend policy optimization to explicitly handle agent heterogeneity, yielding more stable and robust performance across diverse benchmarks. However, these methods continue to rely on conventional entropy regularization, which treats all entropy as uniformly beneficial. In contrast, our approach complements the HARL framework by *denoising stochastic noise in policy entropy*, thereby enabling heterogeneous agents to coordinate exploration more effectively while avoiding instability from spurious entropy fluctuations.

## 3 PRELIMINARIES

### 3.1 PARTIALLY OBSERVABLE DYNAMIC ENVIRONMENT

At time step $t$, an agent $i$ observes its state $s_i$ and select an action $a_i \in \mathcal{A}_i$. After that, the agent $i$ generates its own trajectory $\tau_i$, which could be shared to agents around it. The joint actions of agents $a_t = (a_t^1, \cdots, a_t^N)$ yield the new environment state $s'$ and immediate reward $\{r_t^i\}_{i=1}^N$ according to the transition probability $\mathcal{T} : \mathcal{S} \times \mathcal{A} \times \mathcal{S} \to [0, 1]$ and reward function $\mathcal{S} \times \mathcal{A} \to \mathbb{R}$, respectively, in which $\mathcal{S}$ and $\mathcal{A}$ are the state space and action space of system (i.e. $\mathcal{S} = \prod_{i=1}^M \mathcal{S}_i$ and $\mathcal{A} = \prod_{i=1}^M \mathcal{A}_i$). The main goal of each agent is to learn an optimal policy $\pi^*$ which can maximize the expected return. To get the maximum return, reinforcement learning updates policy model to reduce the loss as following on-policy optimization.

### 3.2 REVISITING ENTROPY REGULARIZATION IN ON-POLICY OPTIMIZATION

The overall policy update is obtained by maximizing a weighted combination of the surrogate objective and the entropy regularization term (Schulman et al., 2017; Yu et al., 2022; Zhong et al., 2024):

$$L^\pi(\theta) = L^{\text{CLIP}}(\theta) + c_{\text{entropy}} L^{\text{entropy}}(\theta) = L^{\text{CLIP}}(\theta) + c_{entropy} \mathbb{E}_t \left[ H(\pi_\theta(\cdot \mid o_t)) \right] \quad (1)$$

where $c_{\text{entropy}}$ is a coefficient controlling the strength of the entropy regularization and $H(\pi_\theta)$ denotes the entropy of the action distribution. $c_{entropy}$ is determined as a hyperparameter. However, in general environments, the mean and variance of $H(\pi_\theta)$ can vary significantly across tasks, making a fixed coefficient suboptimal and potentially leading to either excessive randomness or insufficient exploration.

While Equation (1) stabilizes updates and encourages exploration, it does not distinguish between *useful exploratory entropy* and *detrimental stochastic noise* in the policy distribution. Our work addresses this limitation by explicitly denoising the stochastic noise of policy entropy to achieve more robust optimization in multi-agent settings.

## 4 PROPOSED METHOD

We propose policy entropy manipulation (PEM), an on-policy optimization method for MARL that replaces the raw entropy bonus with a *positive-negative momentum* of policy entropy. The key idea is to interpret policy entropy as a noisy proxy for exploration. Rather than always maximizing it, we *high-pass filter* its temporal variation so that the update uses the *consistent* component of exploration while suppressing spurious fluctuations induced by non-stationarity. This connects to prior work that explicitly controls stochastic gradient noise via momentum (Xie et al., 2021), and extends entropy-regularized on-policy methods (MAPPO (Yu et al., 2022), HAPPO and HAA2C (Zhong et al., 2024)).

### 4.1 IMPROVING GENERALIZATION WITH POSITIVE-NEGATIVE MOMENTUM UNDER STOCHASTIC NOISE

Here, we draw inspiration from a long line of research on stochastic gradient descent (SGD) and stochastic gradient noise (SGN), which has been extensively studied as a means to improve generalization by controlling the stochastic noise component (Hochreiter & Schmidhuber, 1994; Mandt et al., 2017; Smith et al., 2020; Xie et al., 2021). In particular, Xie et al. (2021) introduced *Positive-Negative Momentum (PNM)*, which maintains two approximately independent momentum terms and combines them with positive and negative coefficients. This construction allows the variance of stochastic gradient noise to be explicitly scaled by a factor depending on a tunable parameter. Let $H_t^i \triangleq H\left(\pi_\theta^i(\cdot \mid o_t^i)\right)$ denote the (batch-averaged) entropy of agent $i$'s policy at iteration $t$. Stochastic policy gradient follows:

$$\Delta H_t^i \triangleq H_t^i - H_{t-1}^i + \xi_t^i, \qquad H_{-1}^i \equiv H_0^i \quad (2)$$

where $\xi_t^i$ indicates SGN. Referring to Xie et al. (2021), the key update rule can be expressed as

$$\begin{cases} m_t^{(\text{odd})} = \sum_{j=1,3,\ldots,t}(1-\beta)\beta^{t-j}\Delta H_t^i \\ m_t^{(\text{even})} = \sum_{j=0,2,\ldots,t-1}(1-\beta)\beta^{t-j}\Delta H_t^i \end{cases} \tag{3}$$

$$m_t^i = (1-\beta)\,m_t^{(\text{odd})} + \beta\,m_t^{(\text{even})}, \qquad m_0^i \equiv 0, \quad \beta < 0. \tag{4}$$

where $\beta$ is a momentum parameter $m_t^{(\text{odd})}$ and $m_t^{(\text{even})}$ denote momentum terms updated from odd and even iterations, respectively. The resulting variance of the stochastic noise satisfies

$$\text{Var}(\xi) \approx \left[(1+\beta)^2 + \beta^2\right]\sigma^2 \tag{5}$$

demonstrating that the noise level can be increased or decreased by tuning $\beta$ without altering the expected gradient direction.

This perspective motivates our approach: if policy entropy in reinforcement learning can be regarded as a noisy signal of stochastic exploration, then techniques inspired by momentum-based noise manipulation may provide a pathway to improve generalization performance. In particular, we view the entropy regularization not as a uniformly beneficial signal but as a mixture of informative exploration and detrimental stochastic noise. By drawing on ideas from noise-aware momentum, we aim to design mechanisms that disentangle and suppress the harmful component of entropy in policy optimization.

Intuitively, $m_t^i$ tracks *persistent* trends in entropy while attenuating zero-mean noise. When the environment and co-players are locally stationary, $\mathbb{E}[\Delta H_t^i] \approx 0$, so $m_t^i$ decays; when coordinated exploration ramps up (or collapses), $m_t^i$ responds smoothly without being dominated by transient spikes.

## 4.2 REINFORCEMENT LEARNING OBJECTIVE WITH POLICY ENTROPY MANIPULATION

Let $L^{\text{CLIP}}(\theta)$ be the standard clipped surrogate for the policy actor. Conventional entropy regularization augments this with $c_{\text{entropy}}\,\mathbb{E}_t[H_t^i]$ (Equation. (1)). In the proposed **PEM**, after a one-step warm start, we replace the raw entropy term with the momentum-denoised signal:

$$\textbf{Warm start } (t=0)\text{:} \quad L^\pi(\theta) = L^{\text{CLIP}}(\theta) + c_{\text{entropy}}\,\mathbb{E}_i[H_0^i], \tag{6}$$

$$\textbf{PEM } (t \geq 1)\text{:} \quad L_{\text{PEM}}^\pi(\theta) = L^{\text{CLIP}}(\theta) + \mathbb{E}_i[m_t^i]. \tag{7}$$

In practice, $m_t^i$ is computed from the current batch entropy; the update then backpropagates the loss $-(L_{\text{PEM}}^\pi)$ (equivalently, policy_loss $- m_t^i$ per the implementation) with optional gradient clipping. Setting $\beta{=}0$ recovers a short-memory high-pass filter equal to $\Delta H_t^i$, while omitting $\beta$ entirely (or treating $t{=}0$) reverts to the classical entropy bonus with coefficient $c_{\text{entropy}}$.

**Why it helps.** The momentum on $\Delta H_t^i$ acts as a variance controller on the entropy channel: it damps high-frequency entropy fluctuations that are symptomatic of non-stationarity, while preserving low-frequency, coordinated shifts that represent useful exploration. This yields more stable actor updates, improving generalization across heterogeneous agents and tasks.

## 4.3 MANIPULATION OF STOCHASTIC NOISE IN POLICY ENTROPY

We consider $N$ heterogeneous agents with decentralized policies $\{\pi_{\theta_i}^i(a^i \mid o^i)\}_{i=1}^N$ and a centralized critic $V_\phi(s)$. At timestep $t$, the joint action is $a_t = (a_t^1, \ldots, a_t^N)$ with observations $o_t = (o_t^1, \ldots, o_t^N)$, and $\hat{A}_t$ denotes an advantage estimate (e.g., GAE). Heterogeneous observation/action spaces are allowed; parameter sharing is not assumed.

**Permutation-based surrogate.** Following Zhong et al. (2024), we randomize the order of policy updates using permutations $\sigma \in S_N$. For $i = \sigma(k)$, the incremental ratio is

$$r_t^{i;\sigma}(\theta_i) = \frac{\pi_{\theta_i}^i(a_t^i \mid o_t^i)}{\pi_{\theta_{\text{old}}}^i(a_t^i \mid o_t^i)}, \tag{8}$$

and the cumulative joint ratio up to position $k$ is

$$R_t^{(k;\sigma)}(\theta_{\sigma(1:k)}) = \prod_{j=1}^{k} \frac{\pi_{\theta_{\sigma(j)}}^{\sigma(j)}(a_t^{\sigma(j)} \mid o_t^{\sigma(j)})}{\pi_{\theta_{\text{old}}}^{\sigma(j)}(a_t^{\sigma(j)} \mid o_t^{\sigma(j)})}. \tag{9}$$

The clipped surrogate is then defined as

$$\mathcal{L}_{i|\sigma}^{\text{CLIP}}(\theta_i) = \mathbb{E}_t\Big[\min\big(R_t^{(k;\sigma)}\hat{A}_t, \ \text{clip}(R_t^{(k;\sigma)}, 1-\epsilon, 1+\epsilon)\hat{A}_t\big)\Big], \tag{10}$$

and averaged over permutations $\sigma$.

**Problem with policy entropy.** The standard actor objective augments the surrogate with an entropy bonus $c_{\text{entropy}} \mathbb{E}_t[H(\pi_{\theta_i}^i(\cdot|o_t^i))]$. However, in heterogeneous MARL the entropy $H_t^i$ varies widely across agents and tasks. A fixed weight $c_{\text{entropy}}$ can therefore induce either excessive randomness or insufficient exploration. Moreover, $H_t^i$ is itself a noisy signal under non-stationarity, so blindly maximizing it amplifies stochastic fluctuations.

**Positive–negative momentum for entropy denoising.** To address this, we propose to replace the raw entropy with a *momentum-denoised* variant. Let

$$\Delta H_t^i = H_t^i - H_{t-1}^i, \qquad H_{-1}^i \equiv H_0^i. \tag{11}$$

We construct odd- and even-indexed momentum buffers:

$$m_t^{(\text{odd}),i} = \sum_{j \leq t, \ j \text{ odd}} (1-\beta)\,\beta^{t-j}\,\Delta H_j^i, \qquad m_t^{(\text{even}),i} = \sum_{j \leq t, \ j \text{ even}} (1-\beta)\,\beta^{t-j}\,\Delta H_j^i, \tag{12}$$

and combine them as

$$m_t^i = (1+\beta)\,m_t^{(\text{odd}),i} - \beta\,m_t^{(\text{even}),i}, \qquad m_0^i = 0, \quad \beta \in [0,1). \tag{13}$$

This filter suppresses high-frequency noise in entropy increments while preserving their low-frequency trend, providing a more stable exploration signal.

**Denoised-entropy actor loss.** With a one-step warm start ($t = 0$) we use the standard entropy bonus:

$$L^\pi(\theta) = L^{\text{CLIP}}(\theta) + c_{\text{entropy}}\,\mathbb{E}_i[H_0^i]. \tag{14}$$

For $t \geq 1$, we replace the entropy with the momentum-denoised term:

$$L_{\text{PEM}}^\pi(\theta) = L^{\text{CLIP}}(\theta) + \mathbb{E}_i[m_t^i]. \tag{15}$$

The actor parameters are updated by ascending $L_{\text{PEM}}^\pi(\theta)$ with gradient clipping, while the critic minimizes

$$\mathcal{J}^V(\phi) = \mathbb{E}_t\big[(V_\phi(s_t) - \hat{V}_t)^2\big]. \tag{16}$$

The overall objective is

$$\min_{\theta,\phi}\ \mathcal{J}(\theta,\phi) = -L_{\text{PEM}}^\pi(\theta) + c_v\,\mathcal{J}^V(\phi). \tag{17}$$

**Heuristic Algorithms.** Algorithm 1 summarizes the proposed PEM with positive–negative momentum. The procedure begins with HAPPO-style permutation updates so that heterogeneous agents can be optimized under decentralized policies without assuming parameter sharing. At the first iteration ($t=0$), we employ the conventional entropy regularization term to warm-start the buffer, ensuring that the update is not dominated by transient fluctuations. From $t\geq1$, the entropy increment $\Delta H_t^i = H_t^i - H_{t-1}^i$ is tracked with separate odd and even momentum buffers, which are then combined using a positive–negative momentum rule. This construction suppresses high-frequency noise while preserving low-frequency exploration trends, yielding a denoised entropy signal $m_t^i$. The actor objective replaces the raw entropy bonus with $\mathbb{E}_i[m_t^i]$, so that policy updates are guided by the persistent component of exploration rather than spurious entropy spikes caused by non-stationarity. The critic update remains unchanged. Overall, PEM acts as a variance controller on the exploration channel, reducing instability and "thrashing" while enabling heterogeneous agents to coordinate exploration more effectively.

---

**Algorithm 1** Policy Entropy Manipulation with Positive-Negative Momentum (PEM)

---

**Require:** Decentralized actors $\{\pi^i_{\theta_i}\}^N_{i=1}$, centralized critic $V_\phi$, clip $\epsilon$, entropy coeff. $c_{\text{entropy}}$, momentum $\beta \in [0,1)$, epochs $K$, mini-batches $M$

0: Initialize $\theta, \phi$; set $m^i \leftarrow 0$, $H^i_{\text{prev}} \leftarrow 0$ for all $i$

0: **for** iteration $t = 0, 1, 2, \ldots$ **do**

0:     Collect on-policy trajectories with actors $\{\pi^i_{\theta_i}\}$; compute advantages $\hat{A}_t$ (e.g., GAE)

0:     **for** $k = 1$ to $K$ **do** {PPO-style epochs}

0:         **for** $j = 1$ to $M$ **do** {mini-batches}

0:             Sample a permutation $\sigma \in S_N$; let $i = \sigma(k)$ denote the current agent in the permuted order

0:             Compute cumulative ratio $R^{(k;\sigma)}_t = \prod^k_{u=1} \frac{\pi^{\sigma(u)}_{\theta_{\sigma(u)}}(a^{\sigma(u)}_t | o^{\sigma(u)}_t)}{\pi^{\sigma(u)}_{\theta^{\text{old}}_{\sigma(u)}}(a^{\sigma(u)}_t | o^{\sigma(u)}_t)}$

0:             Clipped surrogate: $\quad L^{\text{CLIP}}(\theta) = \mathbb{E}_t\big[ \min(R^{(k;\sigma)}_t \hat{A}_t, \; \text{clip}(R^{(k;\sigma)}_t, 1-\epsilon, 1+\epsilon)\hat{A}_t)\big]$

0:             Compute per-agent policy entropies $H^i_t = H(\pi^i_{\theta_i}(\cdot | o^i_t))$ (masked means if needed)

0:             **if** $t = 0$ **then** {Warm start}

0:                 $L^\pi(\theta) \leftarrow L^{\text{CLIP}}(\theta) + c_{\text{entropy}} \, \mathbb{E}_i[H^i_t]$

0:             **else**

0:                 $\Delta H^i_t \leftarrow H^i_t - H^i_{\text{prev}}$ for all $i$

0:                 Update odd/even buffers:

$$m^{(\text{odd}),i}_t \leftarrow \sum_{u \le t, \, u \text{ odd}}(1-\beta)\beta^{t-u}\Delta H^i_u, \quad m^{(\text{even}),i}_t \leftarrow \sum_{u \le t, \, u \text{ even}}(1-\beta)\beta^{t-u}\Delta H^i_u$$

0:                 Combine (PNM): $\quad m^i_t \leftarrow (1+\beta)\, m^{(\text{odd}),i}_t - \beta \, m^{(\text{even}),i}_t$

0:                 $L^\pi(\theta) \leftarrow L^{\text{CLIP}}(\theta) + \mathbb{E}_i[m^i_t]$

0:             **end if**

0:             **Actor update:** ascend $L^\pi(\theta)$ with gradient clipping

0:             **Critic update:** minimize $\mathcal{J}^V(\phi) = \mathbb{E}_t[(V_\phi(s_t) - \hat{V}_t)^2]$

0:             $H^i_{\text{prev}} \leftarrow H^i_t$ for all $i$

0:         **end for**

0:     **end for**

0:     $\theta^{\text{old}} \leftarrow \theta$

0: **end for**=0

---

## 5 EXPERIMENTS

This section details the environment, hardware configuration, training parameters, performance metrics, and evaluation methodology employed in our study. Code is available at `https://github.com/anonymous5281/per.git`

### 5.1 ENVIRONMENT

We conduct experiments in two widely used multi-agent reinforcement learning benchmarks that capture complementary aspects of cooperative decision-making: (1) the Multi-Agent Particle Environment-v2 (MPEv2) via PettingZoo, (2) the StarCraft Multi-Agent Challenge v2 (SMACv2), and (3) multi-agent search and rescue using AirSim (Shah et al., 2017). Frameworks instantiate decentralized partially observable Markov decision processes (Dec-POMDPs), providing a controlled yet challenging testbed for evaluating scalability, coordination, and robustness under non-stationarity.

**MPE.** We use the *simple_reference_v2* and *simple_spread_v2* tasks. The former requires agents to reach assigned targets via communication, while the latter demands cooperative landmark coverage with collision avoidance. Agents observe relative positions and velocities, and act through low-dimensional continuous control.

**SMACv2.** SMACv2 extends StarCraft II micromanagement with richer unit compositions and stochasticity. Each agent controls a unit with partial observability, and the team must coordinate to defeat opponents. This benchmark stresses large state spaces, heterogeneous behaviors, and tactical cooperation.

Table 1: Evaluation performance on Spread and Reference scenarios. We report final reward, mean reward, and standard deviation (Std). PEM+HAPPO achieves the best performance in Spread, while MAPPO dominates in Reference.

| Algorithm | Spread | | | Reference | | |
|---|---|---|---|---|---|---|
| | Final | Mean | Std | Final | Mean | Std |
| MAPPO | -76.95 | -83.96 | 7.95 | **-8.88** | -17.34 | 9.30 |
| HAPPO | -61.64 | -71.73 | 10.85 | -9.37 | -20.24 | 10.04 |
| PEM+HAPPO | **-59.78** | -72.42 | 9.69 | -9.17 | -20.88 | 10.21 |
| HAA2C | -106.21 | -104.72 | 4.99 | -34.23 | -38.58 | 4.16 |
| PEM+HAA2C | -102.58 | -104.56 | 5.09 | -30.61 | -37.70 | 3.32 |
| HATRPO | -70.69 | -74.61 | 8.72 | -11.11 | -23.89 | 9.95 |
| MATD3 | -61.08 | -73.12 | 4.20 | -9.54 | 13.39 | 3.42 |
| HATD3 | -63.72 | -74.27 | 3.88 | -9.21 | -13.07 | 3.31 |

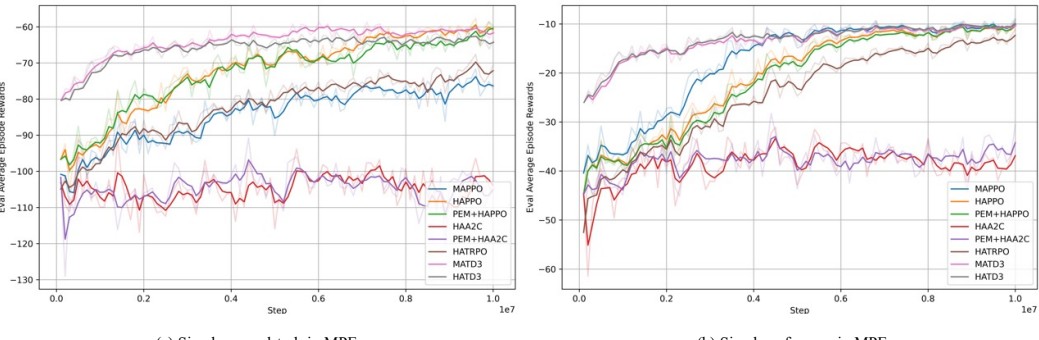

(a) Simple_spread task in MPE

(b) Simple_reference in MPE

Figure 1: Evaluation curves on the Spread scenario. HAPPO significantly outperforms MAPPO, while PEM+HAPPO achieves the best overall performance with improved stability and convergence. In contrast, HAA2C lags far behind, and PEM only provides limited gains, underscoring the importance of robust policy optimization methods in cooperative coordination tasks.

**Search and Rescue.** Multi-UAVs search and rescue environment, a classic control problem, sourced from the Gymnasium reinforcement learning library. We implemented the PEM on MAPPO algorithm, leveraging Ray RLlib Liang et al. (2018) framework for distributed training. For visualization and simulation of environment, we exploited AirSim Shah et al. (2017) with Unreal Engine Epic Games.

## 5.2 PERFORMANCE COMPARISON ON MPEv2

**Simple_Spread_v2.** In the Spread task, performance diverges across algorithm families as shown in Figure 1(a). Among policy-gradient methods, HAPPO markedly outperforms MAPPO (–61.64 vs. –76.95 final reward). The introduction of PEM further enhances results, with PEM+HAPPO achieving the best overall performance (–59.78) and stable learning dynamics. By contrast, HAA2C performs poorly (–106.21), and PEM+HAA2C yields only marginal improvements (–102.58), confirming that simple actor–critic architectures are not competitive in this cooperative coordination setting. HATRPO stabilizes training but remains behind PEM+HAPPO, while MATD3 and HATD3 exhibit intermediate performance with relatively stable but less competitive asymptotic rewards.

**Simple_Reference_v2.** In the Reference task as shown in Figure 1(b), MAPPO remains the top-performing algorithm, with the best final reward (–8.88) and mean performance (–17.34). Both HAPPO (–9.37) and PEM+HAPPO (–9.17) follow closely but do not surpass MAPPO. HAA2C performs poorly (–34.23), though PEM offers some improvement (–30.61).

Among the extended baselines, HATRPO converges moderately (–11.11 final, mean –23.89), while MATD3 (–9.54 final, mean –13.39) and HATD3 (–9.21 final, mean –13.07) achieve competitive results close to MAPPO. Notably, MATD3/HATD3 exhibit low variance, suggesting stable conver-

Table 2: Evaluation performance on Protoss 5v5 and Terran 5v5 with final reward, mean reward, and standard deviation (Std). PEM improves both HAPPO and HAA2C, with environment-dependent effects: incremental gains in Protoss, but substantial improvements in Terran.

|           | Protoss 5v5 |       |      | Terran 5v5 |       |      |
|-----------|-------|-------|------|-------|-------|------|
| Algorithm | Final | Mean  | Std  | Final | Mean  | Std  |
| MAPPO     | 13.83 | 14.63 | 1.53 | 7.77  | 6.98  | 1.08 |
| HAPPO     | 14.28 | 15.32 | 1.92 | 9.21  | 7.36  | 1.12 |
| PEM+HAPPO | **17.07** | 15.49 | 1.88 | **13.20** | 13.14 | 2.15 |
| HAA2C     | 15.58 | 13.48 | 1.79 | 9.27  | 9.98  | 1.59 |
| PEM+HAA2C | 16.84 | 14.26 | 1.87 | 11.95 | 11.91 | 1.64 |

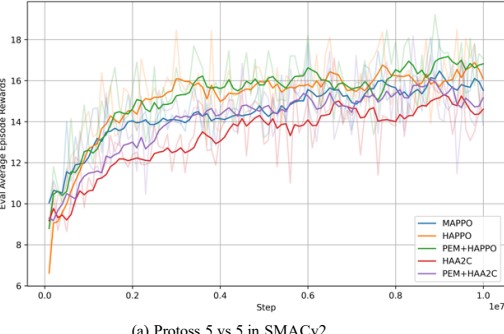 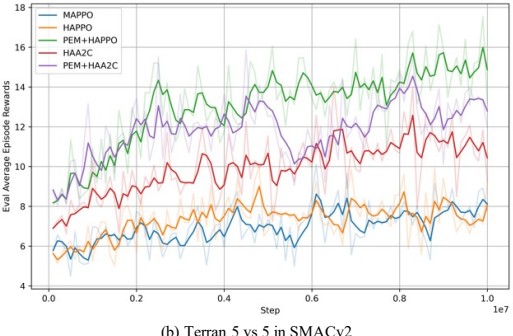

(a) Protoss 5 vs 5 in SMACv2        (b) Terran 5 vs 5 in SMACv2

Figure 2: Evaluation curves on (a) Protoss 5v5 and (b) Terran 5v5. In Protoss, PEM yields moderate gains, with PEM+HAPPO and PEM+HAA2C achieving the best performance. In Terran, PEM is crucial, as PEM+HAPPO and PEM+HAA2C substantially outperform their baselines and MAPPO, highlighting environment-dependent effects of PEM.

gence. These findings confirm that centralized critics (MAPPO) and off-policy actor–critic extensions (MATD3/HATD3) are particularly effective for Simple Reference task.

### 5.3 PERFORMANCE COMPARISON ON SMACV2

We reports the evaluation performance of MAPPO (Yu et al., 2022), HAPPO (Zhong et al., 2024) and HAA2C (Zhong et al., 2024), and the proposed PEM in terms of average episode rewards in evaluation procedure. The learning curves are averaged over evaluation checkpoints, and Table 2 summarizes final and mean performance.

**Protoss 5 vs 5.** Figure 2(a) compares five algorithms on the Protoss 5v5 task. HAPPO achieves stronger performance than MAPPO, reaching a higher final reward (14.28 vs. 13.83). Incorporating PEM yields consistent improvements: PEM+HAPPO attains the best final reward of 17.07, exceeding MAPPO and vanilla HAPPO by +3.2 and +2.8, respectively, while also improving the mean performance. Similarly, PEM+HAA2C (16.84 final) surpasses its baseline HAA2C (15.58), though the margin is less pronounced.

**Terran 5 vs 5.** Figure 2(b) shows the Terran results show a different pattern. PEM+HAPPO achieves the highest final reward (13.20), clearly outperforming both HAPPO (9.21) and MAPPO (7.77). Similarly, PEM+HAA2C (11.95) surpasses its baseline HAA2C (9.27). Unlike in Protoss, where PEM provided incremental improvements, in Terran PEM is crucial for achieving strong performance, especially when combined with HAPPO. This suggests that PEM's effect is environment-dependent, yielding significant gains in Terran where baseline algorithms struggle to converge to high rewards.

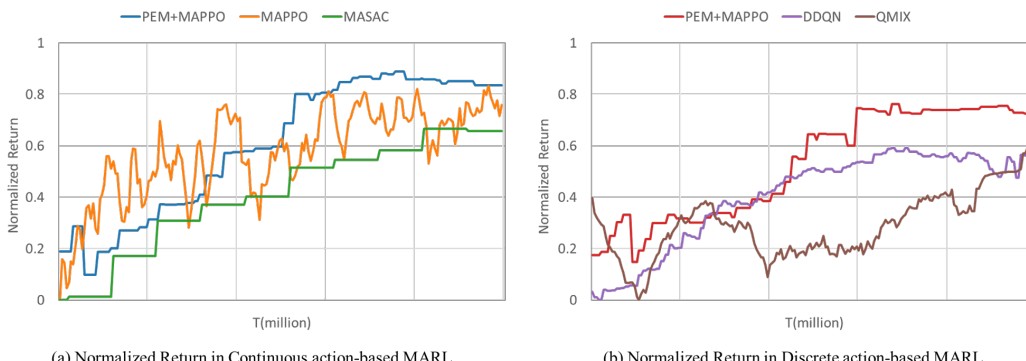

(a) Normalized Return in Continuous action-based MARL      (b) Normalized Return in Discrete action-based MARL

Figure 3: Performance comparison of PEM+MAPPO with MAPPO, MASAC, DDQN, and QMIX in search and rescue environment.

### 5.4 PERFORMANCE COMPARISON ON SEARCH AND RESCUE

In this experiment, we compare continuous and discrete action spaces against standard baselines. For continuous control, we evaluate MAPPO Yu et al. (2022), which extends PPO to multi-agent settings with a centralized critic, and MASAC Lowe et al. (2017), an off-policy maximum entropy method designed for stable and sample-efficient learning. For discrete control, we consider DAD-DQN Van Hasselt et al. (2016) and QMIX Rashid et al. (2020), where QMIX employs a monotonic mixing network for consistent credit assignment.

**Configurations.** All algorithms, including our method, were trained for 200M timesteps with hyperparameters largely following prior work Yu et al. (2022); Lowe et al. (2017); Van Hasselt et al. (2016); Rashid et al. (2020).

**Result.** Figure 3 presents the comparative performance of our proposed EFARL algorithm against several baseline MARL methods, evaluated over 200 million training timesteps. The performance is quantified by the normalized return, where a higher value indicates better performance.

Figure 3(a) illustrates the learning curves for algorithms operating in a continuous action space: PEM+MAPPO, MAPPO, and MASAC. Our proposed PEM method demonstrates a robust and stable learning trajectory, achieving the highest normalized return which consistently stays above 0.8 after 120 million timesteps. In contrast, while MAPPO exhibits rapid initial learning, its performance is marked by high variance and significant oscillations throughout the training process. MASAC shows a more stable learning progression than MAPPO but converges to a suboptimal normalized return of approximately 0.65. This result highlights PEM's superior stability and convergence performance in complex continuous control environments.

Figure 3(b) displays the results for the discrete action space, comparing PEM+MAPPO against DDQN and QMIX. Similar to the continuous setting, PEM achieves the most favorable outcome, reaching a stable normalized return of approximately 0.75 by the 120 million timestep mark. DDQN demonstrates steady learning, gradually improving its performance over time and eventually surpassing QMIX, reaching a return of nearly 0.7 by the end of the training. Conversely, QMIX suffers from significant learning instability; after an initial phase, its performance degrades and remains erratic, ultimately converging to the lowest return among the discrete-action methods.

## 6 CONCLUSION

We introduced Policy Entropy Manipulation (PEM), an on-policy optimization method that denoises entropy signals via positive–negative momentum. By filtering out high-frequency noise while preserving consistent exploration, PEM stabilizes training and improves coordination in heterogeneous multi-agent settings. Experiments on MPEv2, SMACv2, and a multi-agent search-and-rescue task demonstrate that PEM yields smoother learning curves, stronger asymptotic performance, and better generalization than conventional entropy-regularized baselines.

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
