# OpenReview forum: "Revealing Stochastic Noise of Policy Entropy in Reinforcement Learning"
_ICLR.cc/2026/Conference — ICLR 2026 Conference Withdrawn Submission_

### Official Review · Reviewer_VUAe · 2025-10-22

**Soundness:** 3
**Presentation:** 2
**Contribution:** 2
**Rating:** 4
**Confidence:** 2

**Summary:**

This paper attacks On-policy Multi-Agent Reinforcement Learning (MARL). This seeting typically relies on a fixed entropy bonus that conflates useful exploration with stochastic fluctuations from concurrently learning agents. The paper presents Policy Entropy Manipulation
(PEM), a simple, drop-in alternative that treats entropy as a noisy measurement to be denoised rather than uniformly maximized. PEM applies positive–negative momentum to entropy gradient to form a high-pass, variance-controlled signal that preserves persistent exploratory trends while suppressing transient spikes. Experiments demonstrate that PEM yields smoother and more stable learning curves,
and improves coordination under heterogeneous observation/action spaces, resulting in stronger generalization than conventional entropy-regularized baselines.

**Strengths:**

- Identified a neat problem in the Multi-Agent Reinforcement Learning (MARL) setting.

- The solution is simple and effective.

- The evaluation on multiple environments demonstrates the effectiveness of the proposed method.

**Weaknesses:**

- It is difficult to grasp the intuitive reason behind the introduction of the positive-negative momentum of policy entropy. The paper looks like a dedicated engineering effort to outperform existing baselines. However, I have learned relatively little by reading the paper.

- The paper does not offer theoretical guarantees (or justifications) on the effectiveness of their approach.

- The font in Figure 1 is way too small. I also do not understand what the figure is trying to communicate. Similar arguments apply to all other figures.

**Questions:**

- Can the reviewers provide more intuitive rationale behind the introduction of the positive-negative momentum of policy entropy?

- Can we quantify the generalization capability (and learning capability, etc.) theoretically?

- Many claims are confusing. For example, in the abstract, it is argued that smoother and more stable learning curves result in stronger generalization. Why? It seems the two are not logically connected.

---

### Official Review · Reviewer_jRon · 2025-11-01

**Soundness:** 2
**Presentation:** 3
**Contribution:** 2
**Rating:** 2
**Confidence:** 4

**Summary:**

The paper introduces Policy Entropy Manipulation (PEM), an on-policy optimization method for controlling entropy in multi-agent reinforcement learning (MARL). The approach is motivated by the need to distinguish beneficial exploratory signals from detrimental stochastic fluctuations in heterogeneous multi-agent environments. PEM modifies the entropy term by introducing a positive-negative momentum mechanism that aims to suppress high-frequency noise while preserving coordinated, low-frequency exploration. Empirical results are presented across continuous and discrete multi-agent tasks, suggesting that the method stabilizes training dynamics and may improve coordination and generalization among agents.

**Strengths:**

- Addresses a meaningful problem in multi-agent RL regarding entropy-induced instability and coordination.
- Proposes an intuitive mechanism to separate beneficial exploration from random stochastic fluctuations.
- Includes evaluation in both discrete and continuous control environments.
- Presents clear motivation and algorithmic formulation.

**Weaknesses:**

- No theoretical justification for the proposed entropy manipulation mechanism or its claimed benefits.
- Empirical improvements are weak, with performance often comparable to or worse than baseline algorithms.
- Experimental details are incomplete, and unclear metric definitions (e.g., “Final” vs. “Mean”).
- Figures are inconsistently scaled and lack normalization, making direct performance comparison difficult.
- Claims of improved training stability and convergence are not quantitatively supported or measured.

**Questions:**

**Detailed Review:**

The paper proposes Policy Entropy Manipulation (PEM) as a method for controlling policy entropy in multi-agent reinforcement learning. The central idea is that policy entropy contains both useful exploratory signals and harmful stochastic noise. By applying a positive-negative momentum update to the entropy term, PEM aims to suppress spurious fluctuations while preserving coordinated exploration that aids learning stability.

The conceptual motivation is sound, as entropy regulation plays an important role in MARL. However, the method is introduced mainly at an intuitive level, without formal justification or empirical ablation to validate its core assumptions. There is no clear analysis explaining why high-frequency entropy variations are harmful or why the proposed momentum formulation should improve learning dynamics. This absence of theoretical grounding weakens the central argument.

Empirically, the method is evaluated in both discrete and continuous environments, yet the results do not show a consistent advantage over baselines. In many cases, PEM performs similarly to or below the baseline algorithms (HAPPO, MAPPO). For instance, Figure 1 indicates that the PEM-enhanced algorithm tracks the baseline HAPPO curve closely, without clear performance gain or stability improvement. Figures 2 and 3 show similar patterns, where training performance remains nearly identical across methods.

The claim that PEM improves convergence stability is not supported by quantitative metrics, as no explicit stability measurement or variance analysis is provided. It is also unclear how many independent runs were conducted or whether the results were averaged across seeds. This omission limits confidence in the conclusions. The missing environment references (lines 310–323) further complicate reproducibility, and the use of non-normalized scales in figures obscures direct comparison.

While the paper deserves credit for addressing an important issue in MARL and offering a clear algorithmic proposal, the current version lacks sufficient theoretical and empirical depth. The idea could become impactful with further justification, ablation studies, and statistically sound experiments demonstrating consistent gains.

**Questions:**

1. How many independent runs or random seeds were used, and can variance or confidence intervals be reported?
2. What is the difference between “Final” and “Mean” performance values in the tables?
3. How is training stability quantified, and what metric supports the claim of improved convergence?
4. Why does PEM perform similarly to or below HAPPO in several figures, and how should this be interpreted?
5. Could the environment references (lines 310–323) be added for completeness and reproducibility?
6. How does the proposed method handle continuous vs. discrete environments, and why do results appear similar across both settings?

**Details Of Ethics Concerns:**

None.

---

### Official Review · Reviewer_qRsN · 2025-11-01

**Soundness:** 1
**Presentation:** 1
**Contribution:** 1
**Rating:** 0
**Confidence:** 4

**Summary:**

This paper introduces Policy Entropy Manipulation (PEM), a technique to smooth the entropy regularization constraint in multi-agent reinforcement learning (MARL). It replaces the entropy regularization in a regularized objective with positive-negative momentum in the hope of reducing the stochastic noise in the entropy component when optimizing the entropy-regularized objective. Empirically, it investigates the performance of HAPPO and HAAC combined with PEM in a few settings.

**Strengths:**

The paper has the following strength: It comes from an interesting perspective: In deep (multi-agent) reinforcement learning, it is widely believed that training value functions or policies are unstable. Understanding and improving the training dynamic are very valuable. This paper targets the stochastic noise of the entropy term when optimizing an entropy regularized objective, which is an interesting direction to stabilize training and performance.

**Weaknesses:**

The paper has some significant weaknesses and need improvements for further review and publication:
1. It makes weak empirical evaluation and unsubstantiated claims. First, the overall empirical results are weak and not convincing. The paper does not disclose the number of seeds nor include error measures in the plots. In addition, the hypoparameters and the process of tuning them are not included. These missing details invalidate the empirical results, resulting in unsupported claims.
2. Its motivation is not clearly explained. The paper distinguishes useful exploration noise and detrimental stochastic noise throughout, but it didn’t properly define or describe the terms, leaving a hole in its exposition. These terms should be properly discussed in detail either in Sections 3 or 4.
3. Its writing is not polished and has a lot of issues. 1) It is full of typos and errors. See the Questions section for an incomplete list. 2) Its title does not match the content: The paper didn’t really reveal stochastic noise of policy entropy. 3) The related work section is too long, and the discussion of maximum entropy RL and heterogeneous-agent RL are not necessary and distracting. 4) Section 5.1 should also discuss the baselines used. 5) Section 3.2 should be expanded to explain how the objective optimize policies of different agents.

**Questions:**

Questions that may impact the evaluation:
1. Does the proposed method require a buffer/memory that grows linearly with the timestep $t$?
2. Could the authors provide the missing details about the experiments mentioned in the Weaknesses section?

Minor suggestions:
1. Line 112: The mixed use of $s_i$ and $o_i$ throughout the paper creates confusion.
2. Line 126: Inconsistent font in the second equation.
3. Line 169: The grammar of the sentence needs to be fixed.
4. Line 172: The first addition in Eq. 5 should be a subtraction.
5. Line 270: The title should be HAPPO with PEM
6. Lines 450-458: Incorrect citation format.
7. Line 453: DAD-DQN should be a typo.
8. Line 459: “EFARL” should be a typo.

---

### Official Review · Reviewer_shPX · 2025-11-01

**Soundness:** 1
**Presentation:** 2
**Contribution:** 1
**Rating:** 2
**Confidence:** 4

**Summary:**

This work considers entropy not as a universal bonus to be maximized in policy gradient methods, but as a noisy estimate of what should be optimized. It adds a momentum based denoising method to the entropy (summing weighted sums of past entropy changes as an entropy bonus), and adds this as a bonus instead. The paper argues that this perspective is particularly important in multiagent reinforcement learning settings, and performs experiments on the multiagent particle environment, starcraft (Smac v2), and AirSim search and rescue. Their method is called PEM, and they compare it against known methods like MAPPO, HAPPO in combination with them. The results showed that MAPPO remains the best in some tasks (the “Reference” task while PEM+HAPPO slightly improved over HAPPO on the search task), while PEM-based methods slightly improved on some tasks.

**Strengths:**

- The writing was OK, and background and related works were discussed well.

**Weaknesses:**

- The performance was not very strong with inconsistent results.

- I couldn’t find how many runs the experiments were done for? Were these done with a single experimental run? Typically many experimental runs are necessary to obtain statistically significant results.

- There were no error bars.

- I didn’t fully grasp the motivation, why should we consider the entropy as noise that should be denoised? And why is this related to the non-stationarity? Policy entropy can be exactly computed for a given state and policy, so the noise would come mostly from the trajectory sampling. But, I don’t see how you denoise this, other than increasing the sample size.

- The method seemed to be a generic policy gradient algorithm. Why was it used specifically for multi-agent settings? Could the main hypotheses be tested on simpler canonical settings that isolate the claims in the paper?

- Other than reward curves, no other statistics about the performance were provided. This makes it difficult to establish what causes any changes in performance.

**Questions:**

See the weaknesses. I’m afraid it would be difficult for me to change my assessment of this paper, as I believe the quality is not sufficient for me at this stage.

---

### Note · Authors · 2025-11-24

I have read and agree with the venue's withdrawal policy on behalf of myself and my co-authors.